# Recent HIV infections among newly diagnosed individuals living with HIV in rural Lesotho: Secondary data from the VIBRA cluster-randomized trial

**Tsepang Mohloanyane**[1]*, **Dedre Olivier**[1], **Niklaus Daniel Labhardt**[2,3,4], **Alain Amstutz**[2,3,4]

**1** Central University of Technology, Bloemfontein, Free State, South Africa, **2** Division of Clinical Epidemiology, Department of Clinical Research, Basel, Switzerland, **3** University of Basel, Basel, Switzerland, **4** University Hospital Basel, Basel, Switzerland

* tsepimohloa@gmail.com

**Data Availability Statement:** The pseudo-anonymized dataset is available on Zenodo (http://doi.org/10.5281/zenodo.7230264).

## Abstract

### Background

HIV recency assays are used to distinguish recently acquired infection from long-term infection among individuals newly diagnosed with HIV. Since 2015, the World Health Organisation recommends the use of an algorithm to assess recency of infections which is based on an HIV recency assay and viral load (VL) quantification. We determined the proportion of recent HIV infections among participants of the VIBRA (Village-Based Refill of Antiretroviral therapy) cluster-randomized trial in Lesotho and assessed risk factors for these recent infections.

### Methods

The VIBRA trial recruited individuals living with HIV and not taking antiretroviral therapy during a door-to-door HIV testing campaign in two rural districts (Butha-Buthe and Mokhotlong). Samples were collected from participants newly diagnosed and tested for HIV recency using the Asanté HIV-1 Rapid Recency Assay and VL using the Roche Cobas System. Clinical and socio-demographic data were extracted from the trial database. Univariate analysis was conducted to determine factors associated with recent compared to long-term infection.

### Results

Participants were recruited from August 2018 to May 2019 and 184 patient-samples included in this study. The majority were female (108 [59%]) with a median age of 36 years (interquartile range 30–50 years). We found 13 (7.0%) recent infections, while 171 (93.0%) were classified as long-term HIV infections. No conclusive evidence for risk factors of recent infection was found.

### Conclusions

During door-to-door testing among a general population sample in rural Lesotho, 7% of those who were newly diagnosed had acquired HIV in the preceding 6 months. More efforts and research are needed to curb ongoing transmissions in these rural communities.

**Funding:** The study was funded by a Free State Research Grant, obtained by TM, and the Swiss National Science Foundation who funded the overarching VIBRA trial (grants: IZ07Z0_160876/1 & PCEFP3_181355 & 323530_177576). The funders had no role in study design, data collection and analysis, decision to publish, or preparation of the manuscript.

**Competing interests:** The authors have declared that no competing interests exist.

## Introduction

In the past decades, progress in curbing the HIV/AIDS epidemic has been achieved [1]. However, the Joint United Nations Programme on HIV/AIDS (UNAIDS) indicates that there are still regions and population groups where new infection rates are rising [1]. The identification of newly infected individuals is important to help HIV prevention programs to determine where interventions are most needed [2]. Asanté HIV-1 Rapid Recency Assay is a rapid in-vitro immunoassay that is designed to differentiate HIV infections in terms of their recency; whether they are long-term infections or recent infections (within the previous 6 months) [3]. During the early period of HIV infection, antibodies usually have low avidity (binding strength), and their avidity increases as the infection progresses. This assay makes use of this principle [4]. Moreover, this assay incorporates a recombinant protein that contains sequences from all major HIV subtypes, thus is applicable for all HIV subtypes [5]. The World Health Organisation (WHO) recommends using the Asanté HIV-1 Rapid Recency Assay in combination with HIV viral load (VL) quantification as a recent infection testing algorithm (RITA). Individuals taking antiretroviral therapy (ART) should not be tested with the Asanté HIV-1 Rapid Recency Assay, as a low VL leads to low antibody avidity and thus contributes to misclassification. Hence, the addition of VL in the WHO-recommended RITA [6]. The Asanté HIV-1 Rapid Recency Assay has been validated under controlled conditions not only by the manufacturer but also by independent bodies such as the Center for Disease Control [3, 7, 8]. In sub-Sahara Africa, the assay has successfully been applied in various settings such as Nigeria [9], Malawi [10], Zimbabwe [11], and Lesotho [12]. Despite some concerns with its sensitivity [13], the Asanté HIV-1 Rapid Recency Assay does reach acceptable criteria when combined with VL, as outlined by the Consortium for the Evaluation and Performance of HIV Incidence Assays [14], and was used by major national surveys in southern Africa [15].

HIV continues to cause a substantial disease burden in Lesotho, which has the second highest prevalence of HIV in the world [16]. Recency data may help to inform prevention measures to curb the epidemic, but such data are scarce.

The Village-Based Refill of Antiretroviral Therapy (VIBRA) trial was a cluster-randomized trial, that started with a home-based HIV testing campaign among the general population in rural villages of two districts in Lesotho and was used as a platform for this study [17]. The VIBRA trial investigated village-based ART refill following same-day ART initiation versus clinic-based ART refill among individuals found living with HIV but not on ART during a home-based door-to-door HIV testing campaign and its effect on viral suppression at 1 year [18].

This nested study aimed to determine the proportion of recent HIV infections among newly diagnosed participants of the VIBRA study and to assess risk factors for these recent infections.

## Methods

### Study design and participants

This laboratory based cross-sectional study is embedded in VIBRA trial, a cluster-randomized trial that was conducted in two rural districts (Butha-Buthe and Mokhotlong) of Lesotho from 16 August 2018 to 28 May 2019 [18]. Eligible village-clusters were rural, were confined to the catchment area of the 20 health facilities, had a consenting village chief, and had at least 1 registered village health worker who agreed to participate and passed a skill assessment. All community members with a confirmed positive HIV test result (either known HIV-positive or newly tested on the day of the campaign) and not taking ART, i.e., never taken ART (ART-

naive) or had stopped ART more than 30 days prior (ART defaulter), were screened by the study nurses for eligibility. VIBRA trial excluded individuals who planned to get care outside the two study districts (e.g., in neighbouring South Africa), were physically, mentally, and emotionally not able to participate according to the study nurse, were younger than 10 years, had less than 35kg body weight, or were already in care for another chronic disease, because VIBRA trial assessed a village-based ART refill model of care with by that time standard efavirenz-based first-line antiretroviral therapy.

The samples included in this nested study are from consenting and eligible VIBRA trial participants. In addition, we restricted the sample to only those being newly diagnosed on the day of the testing campaign. New diagnosis was self-reported and double-checked by the study staff at the nearby health facility and their HIV testing registry.

## Data collection

If eligible for VIBRA trial, a number of data and laboratory assessments were conducted. After obtaining written informed consent, a venous blood draw as well as a finger prick were done. The finger prick was conducted to perform several point-of-care tests (CD4 cell count, serum creatinine, haemoglobin) to assess parameters for treatment initiation on the day of diagnosis as described in the VIBRA study protocol [17]. In addition, venous blood was drawn by the study nurse for storage of plasma in the -80˚C biobank.

Additionally, a questionnaire was administered to assess clinical, socio-demographic data, which included age, sex, gender, district, employment, schooling and education, regular sexual partner, as well as alcohol and cannabis use among others. Alcohol use was assessed with the CAGE questionnaire which includes four questions that have been proven to assess alcohol dependency [19]. The data were stored in the password protected VIBRA cloud database and could only be accessed through regulated user profiles.

## Laboratory analyses

During door-to-door HIV testing campaign, screening for HIV was performed by using point-of-care Determine HIV1&2 and every reactive test was confirmed with Unigold HIV1&2 [17]. Whole blood samples from new diagnoses were transported in cooler boxes containing ice packs to the nearest hospital laboratory within 30 hours (Mokhotlong hospital laboratory, Butha-Buthe hospital laboratory or Seboche hospital laboratory). On arrival at the laboratories, the Ethylenediamine tetraacetic acid (EDTA) whole blood samples were centrifuged to obtain plasma and were divided into aliquots. At Mokhotlong hospital and Seboche hospital, the plasma was then stored in a -20 degrees celsius freezer and shipped once a week to biobank in the main study laboratory at Butha-Buthe Government Hospital Laboratory. In Butha-Buthe Government Hospital Laboratory the samples were stored in a -80 degrees celsius biobank freezer.

VL quantification was conducted on all the samples collected not only to confirm HIV-1 positive status but to use the VL for the RITA used in this study. This was done at Butha-Buthe Government Hospital Laboratory on the Cobas 4800 system (Roche Molecular Diagnostics [20]). HIV recency testing was conducted using the Asanté HIV-1 Rapid Recency Assay, a point of care test [21]. For this, plasma was pipetted into a separate tube containing sample buffer. The assay test strip was then inserted into the tube containing the sample-buffer mixture. The mixture was absorbed into the absorbent pad of the test strip. Results were read visually after 20 minutes. The final classification of recent versus long-term infection was based on the RITA recommended by the manufacturer and WHO, i.e. incorporating HIV VL [6, 21]. Samples that tested recent with the Asanté HIV-1 Rapid Recency Assay and had a VL greater

than 1,000 copies/ml were classified as true recent infection. Samples than were tested as recent on the Asanté HIV-1 Rapid Recency Assay but had a VL that was equal or less than 1,000 copies/ml were re-classified as long-term infections [6, 21].

## Statistical analysis

Appropriate descriptive statistics were used, i.e. absolute and relative frequencies for categorical data and medians and interquartile ranges for continuous variables. Inferential statistical testing was performed to assess associated factors between clinical and socio-demographic characteristics and recent HIV diagnoses. For this risk factor analysis, we used a logistic regression model. We assessed sex, age, district, schooling, information about sexual partner frequency, substance abuse, CD4 cell count and viral load information in univariate models and in multivariate models as a second step if any associations shown. The choice of these variables was based on known or plausible setting-specific and clinical associations with recent or long-term infection [22]. Results are presented as odds ratios with 95% confidence intervals. All analyses were performed using Stata (version 14, Stata Corporation, Austin/Texas, USA). For all tests, we used complete case analysis, and two-sided p-values with alpha 0.05 level of significance.

## Ethics statement

The VIBRA trial with its nested sub-studies has been approved by the National Health Research and Ethics Committee of the Ministry of Health of Lesotho (ID06–2018) and the ethics committee in Switzerland (Ethikkomission Nordwest- und Zentralschweiz; ID2018–00283). After initial approval, four minor amendments to the study protocol were submitted and approved by the National Health Research and Ethics Committee of the Ministry of Health of Lesotho, whereby the last amendment was submitted specifically for approval of this nested study (Protocol version 9). The study nurse obtained written informed consent in Sesotho, the local language. Illiterate participants provided a thumbprint, and a witness older than 21 years, chosen by the participant, co-signed the consent form. For participants aged < 18 years, a literate caregiver older than 21 years provided consent. All samples were assigned unique identification numbers to anonymize the data and protect confidentiality. At no point during the study the original source of the samples, i.e. participant's names, were revealed.

## Results

During the door-to-door HIV testing campaign, 11,291 were screened for VIBRA trial and 292 (2.6%) were found living with HIV (already known or newly diagnosed) (Fig 1). Of the 292 individuals living with HIV, 257 (88.0%) individuals were eligible for and consented to VIBRA trial.

 For this nested study, we had to exclude another 73 (28.4%) participant-samples, 59 (23.0%) because they were not newly diagnosed and 14 (5.4%) because of insufficient sample quality, leaving a total of 184 (71.6%) participant-samples included in this nested study from the overall VIBRA trial participants.

## Study population characteristics

The demographic, socio-demographic and clinical characteristics of the 184 study participants are shown in Table 1 below. The majority of the participants were female (108/184 [598%]) with a median age of 36 years (interquartile range [IQR] 30–50 years). New infections occurred not only among people below the age of 40 years, but considerable numbers were

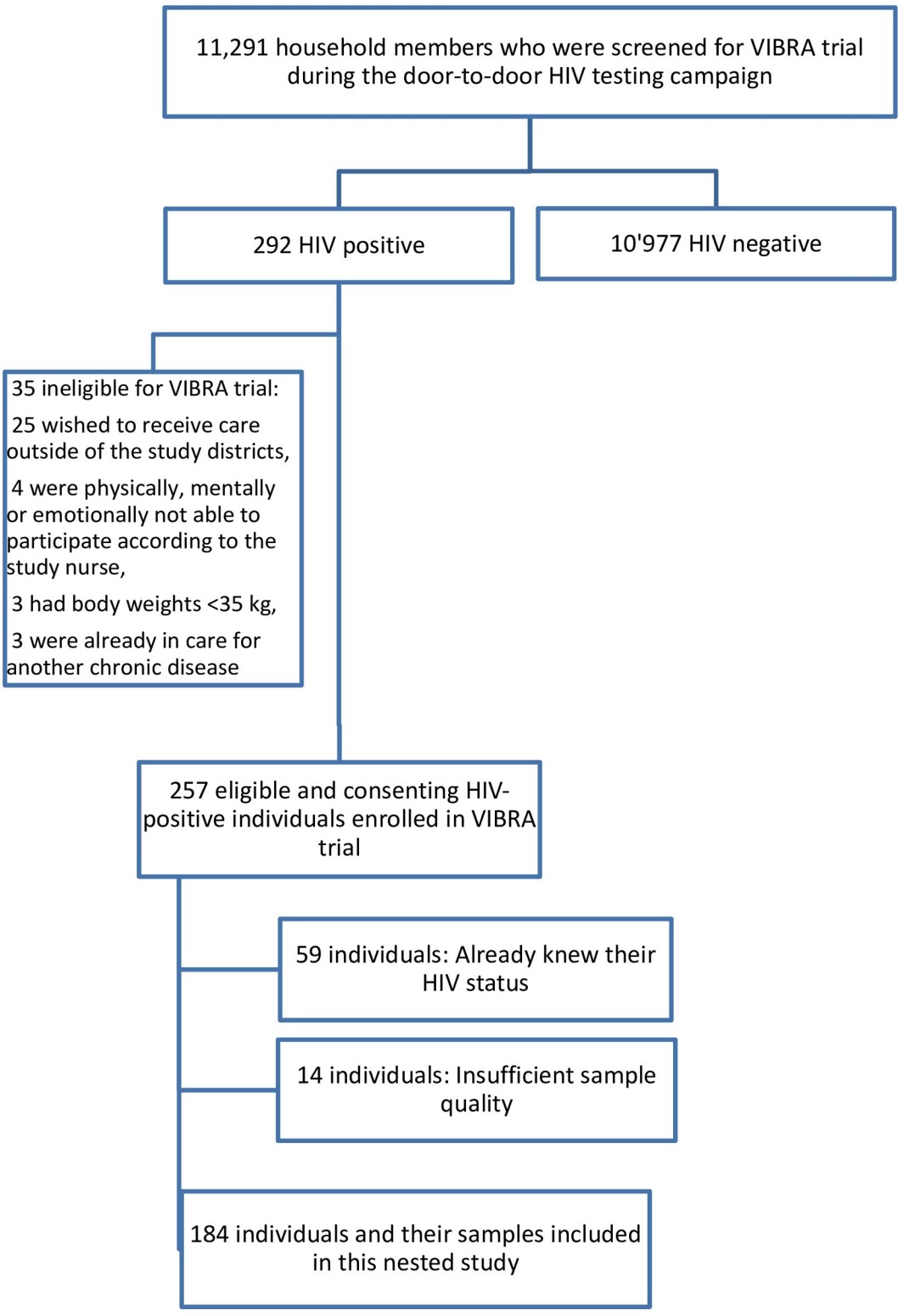

**Fig 1. Study sample flow.**

**Table 1. Demographic and clinical characteristics of study participants, by recency outcome.**

| | Total (n = 184) | Longterm (n = 171) | Recent (n = 13) |
|---|---|---|---|
| Sex | | | |
| Female | 108 (59%) | 97 (57%) | 11 (85%) |
| Male | 76 (41%) | 74 (43%) | 2 (15%) |
| Age in years, median (IQR) | 36 (30–50) | 36 (30–49) | 34 (28–53) |
| Age groups | | | |
| 16–29 years | 45 (24%) | 40 (23%) | 5 (38%) |
| 30–39 years | 60 (33%) | 58 (34%) | 2 (15%) |
| 40–49 years | 33 (18%) | 31 (18%) | 2 (15%) |
| 50–81 years | 46 (25%) | 42 (25%) | 4 (31%) |
| District | | | |
| Butha-Buthe | 16 (9%) | 15 (9%) | 1 (8%) |
| Mokhotlong | 168 (91%) | 156 (91%) | 12 (92%) |
| Schooling in years, median (IQR) | 5 (2–8) | 5 (2–7) | 7 (3–10) |
| 7 years or more of schooling | | | |
| No | 108 (59%) | 103 (60%) | 5 (38%) |
| Yes | 75 (41%) | 67 (39%) | 8 (62%) |
| Missing | 1 (1%) | 1 (1%) | 0 (0%) |
| Education | | | |
| Secondary or higher education | 40 (22%) | 35 (20%) | 5 (38%) |
| No schooling or only primary | 143 (78%) | 135 (79%) | 8 (62%) |
| Missing | 1 (1%) | 1 (1%) | 0 (0%) |
| Employment | | | |
| No employment and no regular income | 155 (84%) | 142 (83%) | 13 (100%) |
| (Self-)Employment with regular income | 29 (16%) | 29 (17%) | 0 (0%) |
| Alcohol abuse | | | |
| No | 44 (24%) | 43 (25%) | 1 (8%) |
| Yes | 15 (8%) | 15 (9%) | 0 (0%) |
| Missing | 125 (68%) | 113 (66%) | 12 (92%) |
| History of local cannabis use | | | |
| No cannabis use | 160 (87%) | 147 (86%) | 13 (100%) |
| Cannabis use | 24 (13%) | 24 (14%) | 0 (0%) |
| Regular sexual partner | | | |
| Yes, one | 90 (49%) | 83 (49%) | 7 (54%) |
| Yes, several | 16 (9%) | 16 (9%) | 0 (0%) |
| No | 72 (39%) | 68 (40%) | 4 (31%) |
| Refused to answer or not applicable | 6 (3%) | 4 (2%) | 2 (15%) |
| CD4 count (cells/µl), median (IQR) | 414 (264–534) | 418 (258–528) | 368 (293–552) |
| CD4 count (cells/µl) groups | | | |
| <200 | 20 (11%) | 20 (12%) | 0 (0%) |
| 200–499 | 81 (44%) | 73 (43%) | 8 (62%) |
| >500 | 47 (26%) | 43 (25%) | 4 (31%) |
| Missing | 36 (20%) | 35 (20%) | 1 (8%) |
| Viral load (copies/mL), log-transformed, median (IQR) | 11 (9–12) | 11 (9–12) | 11 (10–12) |
| Viral load above 30,000 copies/mL | | | |
| No | 83 (45%) | 78 (46%) | 5 (38%) |
| Yes | 101 (55%) | 93 (54%) | 8 (62%) |

Abbreviations: IQR (interquartile range)

Numbers are presented as N (%), unless otherwise stated

also found among people above 50 years. Majority of the participants were from Mokhotlong district (168/184 [91%]).

Lack of employment with no regular income was high (155/184 [84%]). Overall, 143/184 (78%) participants completed primary school only or had no schooling at all, and 108/184 (59%) participants had less than 7 years of schooling. Only 40/184 (22%) participants had secondary or higher education. A few participants reported to have several regular sexual partners (16/184 [9%]), 15/184 (8%) participants screened positive for alcohol dependency and 24/184 (13%) participants had a history of regular cannabis use. Median CD4 cell count was 414 cells/μL (IQR 264–534 cells/μL). All participants had a viral load available, median viral load was 36,650 copies/mL (IQR 8,565–115,500 copies/mL).

## Recency outcomes

Fig 2 outlines the outcomes along the RITA based on the Asanté recency test and HIV VL. Of the 184 included participant-samples, 19 (10.3%) samples were defined as preliminary recent, 6 of those samples (32.6%) were then re-classified as final long-term because their viral load was below 1,000 copies/mL. Overall, among the 184 samples included in this study, we found 13/184 (7.0%) recent infections, i.e., acquired within the previous 6 months, while 171/184 (93.0%) were classified as long-term HIV infections.

## Association of study participants' characteristics and recency

Table 2 displays the univariate logistic regression that displays the associations between participants' characteristics and recency of HIV infection. We were not able to include alcohol nor cannabis use due to low number of events, i.e., recent infection. Across the univariate logistic

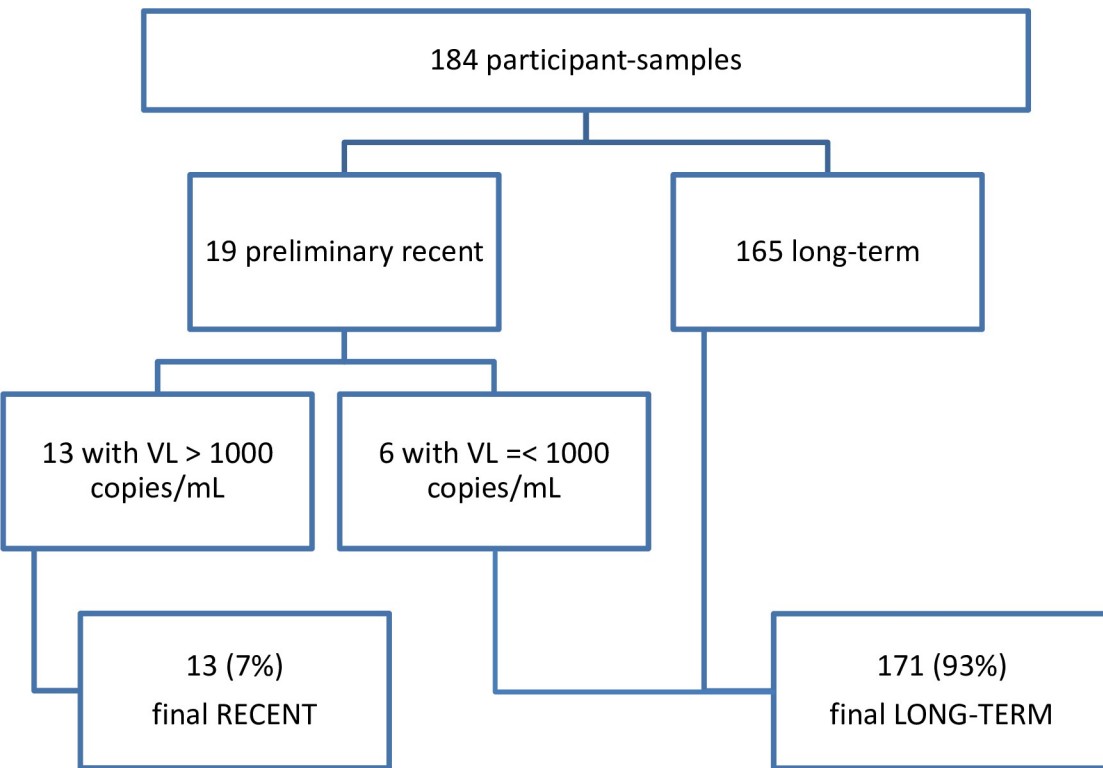

**Fig 2. Recency outcomes using Asanté recency testing and HIV viral load.** Footnote: Abbreviations: VL (viral load).

**Table 2. Univariate logistic regression on recent vs. long-term HIV infection.**

| Characteristic | Recent infection | Univariate logistic regression | |
| --- | --- | --- | --- |
| | | Odds Ratio (95% CI) | P value |
| Female vs Male, n (%) | 11/108 (10.2) vs 2/76 (2.6) | 4.20 (0.90–19.51) | 0.067 |
| Age in years, median (IQR) | 34 (28–53) | 1.00 (0.95–1.04) | 0.687 |
| Butha-Buthe vs Mokhotlong, n (%) | 1/16 (6.3) vs 12/168 (7.1) | 0.87 (0.11–7.13) | 0.894 |
| Schooling in years, median (IQR) | 7 (3–10) | 1.08 (0.94–1.25) | 0.287 |
| No schooling/only primary vs secondary/higher education, n (%) | 8/143 (5.6) vs 5/40 (12.5) | 0.41 (0.13–1.35) | 0.143 |
| Any regular sexual partner vs none or refused to answer, n (%) | 7/106 (6.6) vs 6/78 (7.7) | 0.85 (0.27–2.63) | 0.776 |
| CD4 count in cells/µl, median (IQR) | 368 (293–552) | 1.40 (0.50–3.94) | 0.522 |
| Viral load in copies/mL, median (IQR) | 37,100 (16,700–179,000) | 1.05 (0.80–1.37) | 0.720 |

Abbreviations: IQR (interquartile range), CI (confidence interval)

regression analyses, we were not able to identify significant associations and, thus, no multivariate logistic regression models were built.

## Discussion

This secondary analysis of baseline samples from the VIBRA trial assessed the proportion of recently acquired HIV infections among individuals newly diagnosed with HIV during a door-to-door HIV testing campaign among the general population in 2018/2019 in two rural districts of Lesotho. We found that 7% of new diagnoses were recent infections, i.e., acquired within the 6 months before the testing campaign. Most of the new and recent infections were among women in Mokhotlong district.

No clear evidence for variables associated with recent infections were found. Other similar studies from Eastern Africa and Southeast Asia identified female sex, being married, higher number of sex partners, history of sexually transmitted diseases, younger age and lack of male circumcision as important risk factors, while education and substance use showed no association [22–24]. Due to the low numbers in our sample we lacked statistical power to reach conclusive evidence. Nevertheless, our findings identify ongoing recent transmission of HIV infections, especially in the district of Mokhotlong, with high VL levels of a median of 37,100 copies/mL (IQR 16,700–179,000). All of these new infections occurred among a general population in rural areas that was newly diagnosed during a door-to-door campaign. In a country like Lesotho where HIV prevention, testing and care is a major pillar of the Ministry of Health programme and receives large funding, it is surprising to still find so many new and recent diagnoses during a home-based testing campaign. On the other hand, the remoteness of the study area, especially Mokhotlong district (one of the two mountainous districts in the country with weak infrastructure), may be an important contributor.

A recency study, conducted on 1,025 stored samples of individuals living with HIV from a population survey in 2007 in Kenya, found 6.2% recent infections [23]. However, their denominator included all HIV-positive participant-samples including those with known HIV diagnosis and those on ART and therefore cannot be directly compared to our findings. Another study from Kenya and Zimbabwe, in 2018, was conducted in HIV routine service settings and thus only included new diagnoses. The investigators assessed recency data among female sex workers (FSW), women attending antenatal care (ANC) and general HIV testing facilities. They found similar recency rates among the general testing clients, but lower rates among the

ANC clients and higher rates among the FSW [25]. During a household HIV testing survey in 2017 in Nigeria, 2.9% (11/370) were identified as recent using the same recency algorithm as we did [9]. Again, they included all HIV-positive participant-samples, not only new diagnoses, which is reflected by nearly 20% of the 370 participants being virally suppressed, and thus resulting in an overall lower recency rate. Higher rates of recency were reported in a study conducted in Malawi in 2017/18 among adolescent women aged 15–24 years old [26]. Among the 589 study participants, 11.5% had a recent infection. Young women in sub-Sahara Africa are the population group with the highest HIV incidence [27] and thus higher recency rates than in our general population sample may be expected.

These figures provide important insights into ongoing transmission patterns in the region. However, they should be viewed critically since there is still a lack of proper field validation of these rapid recency assays [28] and especially misleading when not combined with VL testing [29]. More research is needed to validate such assays in the field.

Our study has several limitations. First, a point-of-care rapid recency test and not a laboratory-based recency test was used. Second, this study was conducted in only two districts of Lesotho (Butha-Buthe and Mokhotlong). As a result, the findings of this study may not reflect the characteristics of other districts in Lesotho. Third, the sample size was small. This prevented us to conclude on the risk factor analyses. Fourth, we were not able to get a reliable incidence estimation, as we would have needed a larger sample and preferably from a large-scale cross-section survey, not from a trial. Lastly, we did not measure the presence of antiretrovirals to exclude participants on ART from the recency sample. However, all samples were tested for VL, that improves the false positive recent rate [29], and the VIBRA trial team verified every new diagnosis they found in the field with the surrounding health facilities, thus minimizing the risk of including persons who were already taking ART or knew their status.

Our study found that 7% of new HIV diagnoses found during door-to-door testing in two rural districts of Northern Lesotho were acquired only during the past 6 months. Despite substantial activities of the local HIV programme, recent new infections are still occurring. This indicates ongoing transmission within the general population in these areas in Lesotho. More resources and research is needed in these rural districts to understand the pattern of new transmissions and to tackle them adequately.

## Acknowledgments

We gratefully acknowledge members of the Butha-Buthe Laboratory Hospital team for providing their contribution to this study as well as the VIBRA study team. Most importantly, we thank the study participants.

## Author Contributions

**Conceptualization:** Tsepang Mohloanyane, Dedre Olivier, Niklaus Daniel Labhardt, Alain Amstutz.

**Data curation:** Tsepang Mohloanyane.

**Formal analysis:** Tsepang Mohloanyane, Alain Amstutz.

**Funding acquisition:** Tsepang Mohloanyane, Niklaus Daniel Labhardt, Alain Amstutz.

**Investigation:** Tsepang Mohloanyane.

**Methodology:** Tsepang Mohloanyane, Dedre Olivier, Niklaus Daniel Labhardt, Alain Amstutz.

**Project administration:** Tsepang Mohloanyane, Dedre Olivier.

**Resources:** Alain Amstutz.

**Supervision:** Dedre Olivier, Niklaus Daniel Labhardt, Alain Amstutz.

**Writing – original draft:** Tsepang Mohloanyane.

**Writing – review & editing:** Dedre Olivier, Niklaus Daniel Labhardt, Alain Amstutz.

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
