## [Decision Letter · Decision Letter 0]

10 Jul 2022

PONE-D-22-02237The recency of newly diagnosed HIV infections among the rural general population in Lesotho: Secondary data from the VIBRA cluster-randomized trialPLOS ONE

Dear Dr. Mohloanyane,

Thank you for submitting your manuscript to PLOS ONE. After careful consideration, we feel that it has merit but does not fully meet PLOS ONE’s publication criteria as it currently stands. Therefore, we invite you to submit a revised version of the manuscript that addresses the points raised during the review process.

All three reviewers have carefully reviewed your manuscript and have provided really constructive comments. I would encourage you to address these points carefully if you choose to revise the manuscript. In particular, I draw your attention to two key points raised by Reviewer 3: why the participants who were already aware of their HIV status were not excluded from this analysis, and why the point-of-care recency assay was used rather than a more robust laboratory-based assay. I would also encourage you to develop the discussion to provide stronger interpretation of what these results mean in the broader context of the Lesotho HIV epidemic, and what implications your findings have for the public health response to HIV. =============================

We look forward to receiving your revised manuscript.

Kind regards,

Richard John Lessells, BSc, MBChB, MRCP, DTM&H, DipHIVMed, PhD

Academic Editor

PLOS ONE

Journal Requirements:

Reviewers' comments:

Reviewer's Responses to Questions

**Comments to the Author**

1. Is the manuscript technically sound, and do the data support the conclusions?

Reviewer #1: Partly

Reviewer #2: Yes

Reviewer #3: No

2. Has the statistical analysis been performed appropriately and rigorously? 

Reviewer #1: No

Reviewer #2: Yes

Reviewer #3: No

3. Have the authors made all data underlying the findings in their manuscript fully available?

Reviewer #1: No

Reviewer #2: No

Reviewer #3: Yes

4. Is the manuscript presented in an intelligible fashion and written in standard English?

Reviewer #1: Yes

Reviewer #2: Yes

Reviewer #3: Yes

5. Review Comments to the Author

Reviewer #1: Thank you for the opportunity to review this manuscript addressing the issue of prevalence/ burden of recent infection (<6 months) among newly diagnosed HIV positives individuals taking part in the VIBRA trial. The manuscript was generally well written and easy to follow. However there were some key pieces of information that were not presented/ described and would be helpful for the reader if they were. It would have been helpful if the manuscript included line numbers. These would have made reviewing easier. I have made specific comments on these below

Abstract

Background-

• should be WHO recommends vs advises

• how was newly diagnosed defined

Methods

• Would be helpful to include the inclusion/exclusion criteria for the VIBRA trial. Did the trial only enrol newly diagnosed HIV positives and therefore these were the ones available for recency testing?

Results

• From the results, it seems there were other people other than the newly diagnosed tested for HIV but not for recent infection. Who were these people and why were they not used as a comparison group in the risk factor analysis? One could compare HIV positive recent to HIV positive long term to HIV negative. I could argue that in terms of understanding factors associated with new infections the ideal comparison should be the HIV negatives rather than the HIV positives with long term infection

Conclusion

• The 7.4% recent infection isn't 7.4% of all HIV infections but just of the newly diagnosed ones. What threshold of the proportion of recent infection would be considered high/ significant

Manuscript text

Background/ introduction

• Paragraph 1: comment on the ASANTE assay's performance in determining recent infections under different conditions

• Paragraph 2: the VIBRA trial assessed the effect of village based ART refill on what outcomes/endpoints

• Paragraph 3: what aspect of recent infections is this sub-study evaluating determining – e.g. prevalence, proportion of all new cases etc.

Methods

Study design and participants

• What outcomes were measured in the VIBRA trials

• What was the eligibility criteria for the VIBRA trial and for this nested study? Were they the same?

Data collection

• What was the eligibility criteria for the VIBRA trial? What were the eligibility criteria for this nested study/ sub-study

Laboratory analyses

• add details regarding Asante's performance in determining recency of infection under different conditions

Statistical analysis

• The authors wrote that the multivariable logistic regression model was fitted with the two most important clinical factors (age and gender). Why only these two variables? Please discuss how multivariable model building was done

Results

• Was there any data collected on the HIV negatives? From the results, it does not seem so. This is why it was important to have included the eligibility criteria for this current study in the methods

Study population characteristics

• The authors wrote “Majority of the participants were from Mokhotlong district (216/243 [89%]) and 13/18 (72%) participants were located in the catchment area of the same 5 out of 19 facilities”. This doesn’t make sense to me. Please check

• In this same paragraph, the authors wrote “Most participants were unaware of their HIV-positive status (184/243 [76%]).” This doesn’t make sense as only participants with unknown HIV status were tested in the first place

Table 2:

• Please add a column with data on the prevalence of the outcome presented as n/N (%)

• There is no need to present the β coefficients in the table

• Since the majority of new infections were from women, was there an attempt to look at associated factors among women only

Discussion

• Paragraph 1: What does it mean that only gender was associated with recent infection and what are the implications for HIV prevention

• Paragraph 2: wasn’t the denominator in the Kenyan study all HIV positives and not just newly diagnosed. Please check tht you are comparing like with like

Reviewer #2: This study first identified recent HIV infections among the Village-Based Refill of Antiretroviral Therapy (VIBRA) study population, and then described and assessed risk factors for these recent infections. The study design is good and the results are instructive. There are some statistical details that need to be revised—

1) In abstract, “…, only female gender was predictive for a recent HIV infection”— although in a regression model we call the independent variables predictors, it should be careful to state that one variable is predictive for another variable because only association is tested, and there is no a predication model built or prediction study performed.

2) There were 13/18 (72%) participants located in the catchment area of the same 5 out of 19 facilities. It is implied that there may be significant ongoing transmission. What’s the chance of observing 13/18 in 5/19 or more extreme situations? Please calculate the p-value here.

3) I suggest including Age and Sex in all the regression models.

Reviewer #3: Many thanks for presenting me with this opportunity to review this paper. In the paper, the authors describe a study where they aimed to determine recent HIV infections among a study population in Lesotho and describe and assess risk factors for these.

Below I raise a number of points / questions for consideration which, I believe, if addressed will strengthen the paper. I have a number of major concerns with the analyses, the main one of which is that “newly” diagnosed people who were already aware of their status and were not “new” were included in the recency analysis – such individuals must be excluded from recency testing.

Title

1. The title is a little misleading as the authors are not measuring the recency of newly diagnosed infections but of the infections themselves – these are different things – I suggest amending to make clear they are conducting surveillance of recency HIV infections and not of diagnoses (which would require measurement from estimated sero-conversion to date of diagnosis).

Abstract

2. Best to describe sample as pertaining to people newly diagnosed rather than “samples of new HIV diagnoses” as the sample does not pertain to a new diagnosis.

3. If these are all people newly diagnosed with HIV how could any be aware of their HIV status? If they were aware then by definition they are not newly diagnosed with HIV and should not be in the sample.

4. Given the test is binary (it either is or is not recent) and the people testing are actual people, what do confidence intervals add / how are they to be interpreted?

5. Suggest remove “Although some characteristics showed a trend” as this adds little.

6. I would argue that finding 18 people who probably acquired their infection in the past 6 months does not imply “significant ongoing transmission in these rural communities” – please qualify or revise / remove.

Introduction

7. The opening sentence is vague as states “past decades” and “massive progress” – I would argue that progress has been modest depending on the setting – massive progress would appear to be contrary to the conclusion that there is significant ongoing transmission – please quantify / amend.

8. There is a large body of literature to reference re. the use of these assays in southern Africa – there should be some discussion relating to this to set context – the introduction is currently light.

Methods

9. CD4 cell test mentioned as is alcohol and cannabis use (alcohol use described in detail) but these factors are hardly deliberated on in the results and not at all in the discussion. Having included such variables in their analysis, it would be helpful for the authors to make clear their a-priori hypotheses how such factors may influence whether an infection is recent or not and then these hypotheses deliberated on in their discussion.

10. Tiny thing but data usually = plural so data were stored rather than was stored.

11. It would be helpful to reference some of the diagnostics.

12. May be preferrable to state that tests were conducted rather than done.

13. Can the authors please make clear why a rapid recency assay was used (which is documented as having numerous limitations – see following links) rather than a laboratory-based recency assay (that also have limitations but fewer than the rapid assay) – this is an important point given what we kknow about the performance of the rapid test and given the samples I believe were tested in a laboratory and not in the field - https://publichealth.jmir.org/2022/3/e34410 & https://pubmed.ncbi.nlm.nih.gov/33499732/

Results

14. Could the authors revise how they present their results – for example the sentence “314 among them were HIV-positive on the day of enrolment and 257 of them were enrolled into VIBRA trial.” Starts with a number, refers to people as “them” and only present n and not also % - including %s would be helpful throughout – sometimes the authors present n, sometimes N, and sometimes % - please can the authors standardise and improve their presentation of results.

15. Figure 1 presents numbers with ‘ and not , - I have not seen this before in a paper and would suggest the authors follow generally accepted ways of presenting numbers.

16. Figure 1 does not address the issue raised in the abstract that some of the “newly” diagnosed people were aware of their status.

17. Also be helpful to make clear the 6 VL <1000 transfer into the 225 non-recents in figure 2

18. The authors state that “Majority of the participants were from Mokhotlong district (216/243 [89%]) and 13/18 (72%) participants were located in the catchment area of the same 5 out of 19 facilities.” – as I don’t believe either the 5 or 19 facilities have previously been referred to, it is unclear what facilities the authors are referring to – please clarify

19. The issue of 59 participants being aware of their status is surprising as they should have been excluded from the study sample at the start – the authors need to revise their analytical approach so these individuals are excluded from the start as they does not meet usual inclusion criteria for recency testing. I would also argue that 59 people knowing their status cannot be referred to as “only” given this is a surprisingly high number given they are supposed to be “newly” diagnosed (i.e. the number should be 0).

20. It is unclear why the authors include the 59 people who were aware of their HIV status in subsequent analysis – they need to be removed as they are not newly diagnosed and therefore do not qualify for recency testing.

21. The authors present low values (including 1 and 0) by very specific and potentially defining variable categories in their table 1 – I personally would mask facility names and not present small values (instead adopting e.g. <5) – I feel how the data are presented is problematic.

22. Confidence intervals are presented alongside definitive results where both the n and the N are certain – could the authors please make clear how we are to interpret these confidence intervals.

Discussion

24. The first paragraph of the discussion is almost written like a bullet point summary of the results and presents the main findings again flatly – it would be helpful if the finding of being female being associated with recent infection was explained for example.

25. Good to see the authors share limitations – these must also include the fact people who new their status were included in the sample and that a rapid test was used rather than a laboratory based LAg assay.

26. The discussion lacks a so-what – next steps – what this means. The discussion needs to be greatly strengthened.

6. PLOS authors have the option to publish the peer review history of their article (what does this mean?). If published, this will include your full peer review and any attached files.

Reviewer #1: **Yes: **Tendesayi Kufa

Reviewer #2: No

Reviewer #3: **Yes: **Brian Rice

---

## [Author Response · Author response to Decision Letter 0]

6 Sep 2022

I have uploaded the response letter also as a separate attachment. Formatting in the attachment may make it easier to follow the response point by point than here.

Response to Reviewers

The recency of newly diagnosed HIV infections among the rural general population in Lesotho: Secondary data from the VIBRA cluster-randomized trial

Journal Requirements:

Response: We updated our manuscript to meet PLOS ONE’s style requirements

Response: We are already preparing the anonymized dataset for upload onto Zenodo and will be ready with the DOI in time if the manuscript is accepted for publication.

Reviewer #1: 

1. Thank you for the opportunity to review this manuscript addressing the issue of prevalence/ burden of recent infection (<6 months) among newly diagnosed HIV positives individuals taking part in the VIBRA trial. The manuscript was generally well written and easy to follow. However there were some key pieces of information that were not presented/ described and would be helpful for the reader if they were. It would have been helpful if the manuscript included line numbers. These would have made reviewing easier. I have made specific comments on these below.

Response: Thank you for taking your time to review this manuscript and very grateful for the constructive criticism. We have now included line numbers in the manuscript to make it easier for reviewing. 

2. Abstract / Background

• should be WHO recommends vs advises

Response: We have adapted accordingly

3. • how was newly diagnosed defined

4. Response: Newly diagnosed was defined as people who were not aware of their HIV positive status at the day of diagnosis during the HIV door-to-door testing campaign. This was self-reported information and confirmed by the study staff by systematically searching all HIV testing registries at the nearby health facilities. Obtaining incomplete information may be a limitation (see discussion), but due to the local setting with small rural villages and the close interaction of the study staff with the village health workers and all health facilities and their registries we reduced this bias as much as possible. And then of course, we added viral load testing to reduce false-positive recent infections.

5. Methods

• Would be helpful to include the inclusion/exclusion criteria for the VIBRA trial. Did the trial only enrol newly diagnosed HIV positives and therefore these were the ones available for recency testing?

Response: We included the detailed eligibility criteria of VIBRA trial (on cluster and individual level) in the first paragraph of the Methods in the main text. In the abstract, we also reflected the criteria, but had to shorten them due to word count limitations. 

Thank you for pointing out this important point: In addition, we now restricted the sample to only those that were newly diagnosed. Although other studies (e.g. https://onlinelibrary.wiley.com/doi/10.1002/jia2.25669 or https://journals.plos.org/plosone/article?id=10.1371/journal.pone.0155498) include also those with known diagnoses (or even on ART) and then automatically classify them as “long-term infections”, we agree that they should rather be excluded from the denominator in our case. Thus, all numbers and tables were adapted. This slightly changed the proportion of recent infections from 7.4% down to 7.0% and no conclusive evidence for an association between recent infection and participant characteristics could be established anymore.

6. Results

From the results, it seems there were other people other than the newly diagnosed tested for HIV but not for recent infection. Who were these people and why were they not used as a comparison group in the risk factor analysis? One could compare HIV positive recent to HIV positive long term to HIV negative. I could argue that in terms of understanding factors associated with new infections the ideal comparison should be the HIV negatives rather than the HIV positives with long term infection

Response: We excluded now all those patient-samples that were not newly diagnosed. However, we do suggest keeping the long-term infections as the comparator for recent infections to have the same common denominator (HIV-positive) with the only difference of being recent versus long-term infected – similar to this study (https://pubmed.ncbi.nlm.nih.gov/33653290/). Moreover, we unfortunately do not have the same detailed characteristics for the population that tested HIV-negative (education, schooling, etc).

7. Conclusion

 The 7.4% recent infection isn't 7.4% of all HIV infections but just of the newly diagnosed ones. What threshold of the proportion of recent infection would be considered high/ significant

Response: We modified the sentence and added “7% of newly diagnosed”. Defining a level of significance is difficult, but we provide comparable figures in the discussion. In order to provide a more neutral statement, we deleted the word “significant” and reworded the conclusion, so that we do not imply a judgement from our side. The updated conclusion reads as follows:

During door-to-door testing among a general population sample in rural Lesotho, 7% of newly diagnosed individual living with HIV were acquired less 6 months ago. More efforts and research are needed to curb ongoing transmissions in these rural communities.

8. Manuscript text / Background/ introduction

• Paragraph 1: comment on the ASANTE assay's performance in determining recent infections under different conditions

Response: We have added more information on the performance of Asante in a controlled as well as setting-specific environment. Certainly, the performance of this point-of-care test is still not optimal, when combined with viral load measurement, it reaches adequate performance measures. In detail, we have added the following paragraph in the Introduction:

The Asanté HIV-1 Rapid Recency Assay has been validated under controlled conditions not only by the manufacturer but also by independent bodies such as the Center for Disease Control (3,7,8). In sub-Sahara Africa, the assay has successfully been applied in various settings such as Nigeria (9), Malawi (10), Zimbabwe (11), and Lesotho (12). Despite some concerns with its sensitivity (13), the Asanté HIV-1 Rapid Recency Assay does reach acceptable criteria when combined with VL, as outlined by the Consortium for the Evaluation and Performance of HIV Incidence Assays (14), and was used by major national surveys in southern Africa (15).

9. Paragraph 2: the VIBRA trial assessed the effect of village-based ART refill on what outcomes/endpoints?

Response: The primary endpoint was viral suppression after 1 year. We added more information to this paragraph to clarify as well as in the Methods section.

10. • Paragraph 3: what aspect of recent infections is this sub-study evaluating determining – e.g. prevalence, proportion of all new cases etc.

Response: The study focuses on the proportion of recently acquired HIV infections among newly diagnosed patients from the VIBRA trial participants identified during a door-to-door HIV testing campaign among the general population.

11. Methods / Study design and participants

What outcomes were measured in the VIBRA trials

Response: The primary endpoint was viral suppression after 1 year. We added more information to this paragraph to clarify as well as in the Methods section. However, for more outcomes we would like to refer to the VIBRA trial publications since they do not fall into the scope of this publication.

12. What was the eligibility criteria for the VIBRA trial and for this nested study? Were they the same?

Response: The eligibility criteria of this study was based on the VIBRA trial. The blood samples from eligible and consenting VIBRA participants are the basis for this study. A table showing the eligibility criteria of VIBRA has been inserted in the manuscript

13. Data collection

What were the eligibility criteria for the VIBRA trial? What were the eligibility criteria for this nested study/ sub-study

Response: We included the detailed eligibility criteria of VIBRA trial (on cluster and individual level) in the first paragraph of the Methods. In addition, we now restricted the sample to only those that were newly diagnosed, based on self-reporting.

14. Laboratory analyses

 add details regarding Asante's performance in determining recency of infection under different conditions

Response: We have added substantial literature on the performance of Asante in the literature section (see point 8) and suggest to rather focus on the procedure/handling in this part of the manuscript.

15. Statistical analysis

The authors wrote that the multivariable logistic regression model was fitted with the two most important clinical factors (age and gender). Why only these two variables? Please discuss how multivariable model building was done

Response: Due to lower numbers after changing the sampling frame, we decided to stick to a univariate analysis and only build a multivariate model if there are any significant associations found. We reworded this part in the manuscript: 

We assessed sex, age, district, schooling, information about sexual partner frequency, substance abuse, CD4 cell count and viral load information in univariate models and in multivariate models as a second step if any associations shown. The choice of these variables was based on known or plausible setting-specific and clinical associations with recent or long-term infection (21).

16. Results

Was there any data collected on the HIV negatives? From the results, it does not seem so. This is why it was important to have included the eligibility criteria for this current study in the methods

Response: HIV-negative participants were not included in this study and thus, indeed, we do not have detailed characteristics of these people.

17. Study population characteristics

 The authors wrote “Majority of the participants were from Mokhotlong district (216/243 [89%]) and 13/18 (72%) participants were located in the catchment area of the same 5 out of 19 facilities”. This doesn’t make sense to me. Please check

Response: Thank you for pointing this out. We deleted this misleading sentence and reduced the characteristics table to district only, since the inclusion of facilities does not add much valuable information but rather raises concerns in terms of confidentiality as outlined by reviewer 3.

18. In this same paragraph, the authors wrote “Most participants were unaware of their HIV-positive status (184/243 [76%]).” This doesn’t make sense as only participants with unknown HIV status were tested in the first place

Response: We changed the presented sample, i.e., strictly only newly diagnosed as per self-report and, thus, this statement is not true anymore.

19. Table 2:

Please add a column with data on the prevalence of the outcome presented as n/N (%)

Response: We added the column.

20. There is no need to present the β coefficients in the table

Response: We deleted the β coefficients

• Since the majority of new infections were from women, was there an attempt to look at associated factors among women only

Response: We looked at associated factors among women only but found no associated factors, either (data not shown). We think adding this data will not add much information.

21. Discussion

 Paragraph 1: What does it mean that only gender was associated with recent infection and what are the implications for HIV prevention

Response: Since we changed the sample sightly, we are unfortunately not able to show this association anymore. However, we expanded our discussion with other informative sections and arguments to increase its depth and to be more informative.

22. Paragraph 2: wasn’t the denominator in the Kenyan study all HIV positives and not just newly diagnosed. Please check tht you are comparing like with like

Response: Thank you for pointing this out. We reflected on this in the discussion. The same applies to the study from Nigeria. While these are important recency data, comparing the figures directly with each other (also due to field test performance issues) has to be viewed critically. We added a paragraph on this, too. We hope we could meet the reviewers’ expectations and improve the discussion significantly. 

Reviewer #2: 

1. This study first identified recent HIV infections among the Village-Based Refill of Antiretroviral Therapy (VIBRA) study population, and then described and assessed risk factors for these recent infections. The study design is good and the results are instructive. There are some statistical details that need to be revised

Response: Thank you for your positive feedback.

2. Abstract

“…, only female gender was predictive for a recent HIV infection”— although in a regression model we call the independent variables predictors, it should be careful to state that one variable is predictive for another variable because only association is tested, and there is no a predication model built or prediction study performed.

Response: Thank you for making this important point. We adapted slightly our sample (due to comments from reviewer 1 and 3) and thus our results of the regression model changed, too. We had to restrict our sample further, thus reduced data, and thus lower statistical power. We are now not able anymore to conclude on any significant association between participant characteristics and recency outcome. We reworded all relevant parts in the abstract accordingly.

3. There were 13/18 (72%) participants located in the catchment area of the same 5 out of 19 facilities. It is implied that there may be significant ongoing transmission. What’s the chance of observing 13/18 in 5/19 or more extreme situations? Please calculate the p-value here.

Response: Thank you for pointing this out. We deleted this misleading sentence and reduced the characteristics table to district only, since the inclusion of facilities does not add much valuable information but rather raises concerns in terms of confidentiality as outlined by reviewer 3.

4. I suggest including Age and Sex in all the regression models.

Response: Due to lower numbers after changing the sampling frame, we decided to stick to a univariate analysis and only build a multivariate model if there are any significant associations found. We reworded this part in the manuscript: 

We assessed sex, age, district, schooling, information about sexual partner frequency, substance abuse, CD4 cell count and viral load information in univariate models and in multivariate models as a second step if any associations shown. The choice of these variables was based on known or plausible setting-specific and clinical associations with recent or long-term infection (21).

Reviewer #3: 

1. Many thanks for presenting me with this opportunity to review this paper. In the paper, the authors describe a study where they aimed to determine recent HIV infections among a study population in Lesotho and describe and assess risk factors for these.

Below I raise a number of points / questions for consideration which, I believe, if addressed will strengthen the paper. I have a number of major concerns with the analyses, the main one of which is that “newly” diagnosed people who were already aware of their status and were not “new” were included in the recency analysis – such individuals must be excluded from recency testing.

Response: Thank you for your valuable feedback – you raised a some very important points and we hope we could address them in our amended manuscript. 

We have adjusted our analyses and included now only those that were newly diagnosed (self-reported). Although other studies (e.g. https://onlinelibrary.wiley.com/doi/10.1002/jia2.25669 or https://journals.plos.org/plosone/article?id=10.1371/journal.pone.0155498) include also those with known diagnoses (or even on ART) and then automatically classify them as “long-term infections”, we agree that they should rather be excluded from the denominator. Thus, all numbers and tables were adapted. This slightly changed the proportion of recent infections from 7.4% down to 7.0% and no conclusive evidence for an association between recent infection and participant characteristics could be established anymore. 

2. Title

The title is a little misleading as the authors are not measuring the recency of newly diagnosed infections but of the infections themselves – these are different things – I suggest amending to make clear they are conducting surveillance of recency HIV infections and not of diagnoses (which would require measurement from estimated sero-conversion to date of diagnosis).

Response: We adapted the title to: 

Recent HIV infections among newly diagnosed individuals living with HIV in rural Lesotho: Secondary data from the VIBRA cluster-randomized trial

3. Abstract

Best to describe sample as pertaining to people newly diagnosed rather than “samples of new HIV diagnoses” as the sample does not pertain to a new diagnosis.

Response: We adapted accordingly throughout the abstract.

4. If these are all people newly diagnosed with HIV how could any be aware of their HIV status? If they were aware then by definition they are not newly diagnosed with HIV and should not be in the sample.

Response: As explained in point 1, this has been resolved.

5. Given the test is binary (it either is or is not recent) and the people testing are actual people, what do confidence intervals add / how are they to be interpreted?

Response: We agree to remove the confidence interval since it does not add reliable information

6. Suggest remove “Although some characteristics showed a trend” as this adds little.

Response: We remove this misleading phrase.

7. I would argue that finding 18 people who probably acquired their infection in the past 6 months does not imply “significant ongoing transmission in these rural communities” – please qualify or revise / remove.

Response: In order to provide a more neutral statement, we deleted the word “significant” and reworded the conclusion, so that we do not imply a judgement from our side. The updated conclusion reads as follows:

During door-to-door testing among a general population sample in rural Lesotho, 7% of newly diagnosed individual living with HIV were acquired less 6 months ago. More efforts and research are needed to curb ongoing transmissions in these rural communities.

8. Introduction

The opening sentence is vague as states “past decades” and “massive progress” – I would argue that progress has been modest depending on the setting – massive progress would appear to be contrary to the conclusion that there is significant ongoing transmission – please quantify / amend.

Response: We agree and amended accordingly.

9. There is a large body of literature to reference re. the use of these assays in southern Africa – there should be some discussion relating to this to set context – the introduction is currently light.

Response: We have added more information about the performance of the recency assay in a controlled as well as setting-specific environment. In detail, we have added the following paragraph in the Introduction:

The Asanté HIV-1 Rapid Recency Assay has been validated under controlled conditions not only by the manufacturer but also by independent bodies such as the Center for Disease Control (3,7,8). In sub-Sahara Africa, the assay has successfully been applied in various settings such as Nigeria (9), Malawi (10), Zimbabwe (11), and Lesotho (12). Despite some concerns with its sensitivity (13), the Asanté HIV-1 Rapid Recency Assay does reach acceptable criteria when combined with VL, as outlined by the Consortium for the Evaluation and Performance of HIV Incidence Assays (14), and was used by major national surveys in southern Africa (15).

10. Methods

CD4 cell test mentioned as is alcohol and cannabis use (alcohol use described in detail) but these factors are hardly deliberated on in the results and not at all in the discussion. Having included such variables in their analysis, it would be helpful for the authors to make clear their a-priori hypotheses how such factors may influence whether an infection is recent or not and then these hypotheses deliberated on in their discussion.

Response: Due to lower numbers after changing the sampling frame, we decided to stick to a univariate analysis and only build a multivariate model if there are any significant associations found. We reworded this part in the manuscript: 

We assessed sex, age, district, schooling, information about sexual partner frequency, substance abuse, CD4 cell count and viral load information in univariate models and in multivariate models as a second step if any associations shown. The choice of these variables was based on known or plausible setting-specific and clinical associations with recent or long-term infection (21).

Moreover, we added some more thoughts on the outcome of this updated regression analysis in the discussion:

No clear evidence for variables associated with recent infections were found. Other similar studies from Eastern Africa and Southeast Asia identified female sex, being married, higher number of sex partners, history of sexually transmitted diseases, younger age and lack of male circumcision as important risk factors, while education and substance use showed no association (22–24). Due to the low numbers in our sample we lacked statistical power to reach conclusive evidence.

11. Tiny thing but data usually = plural so data were stored rather than was stored.

Response: We adapted accordingly

12. It would be helpful to reference some of the diagnostics.

Response: We added individual references for all laboratory tests conducted as part of this study; for the other laboratory tests we referred to the VIBRA study protocol for more details.

13. Maybe preferrable to state that tests were conducted rather than done.

Response: We adapted accordingly

14. Can the authors please make clear why a rapid recency assay was used (which is documented as having numerous limitations – see following links) rather than a laboratory-based recency assay (that also have limitations but fewer than the rapid assay) – this is an important point given what we know about the performance of the rapid test and given the samples I believe were tested in a laboratory and not in the field - https://publichealth.jmir.org/2022/3/e34410 & https://pubmed.ncbi.nlm.nih.gov/33499732/

Response: Thank you for this important point. This project was part of a MSc thesis of myself (Ts’epang) and unfortunately we only had limited resources and laboratory test choices available. In Lesotho, at the time, this was exactly the assay used to assess recency in the routine HIV programme as well as in other large-scale national surveys in the region. When combined with viral load testing (as we did), the test performance significantly improves. However, we do agree that it has its limitations. We added and discussed the references you provided. In detail, we added a paragraph and word of caution in the introduction:

Despite some concerns with its sensitivity (13), the Asanté HIV-1 Rapid Recency Assay does reach acceptable criteria when combined with VL, as outlined by the Consortium for the Evaluation and Performance of HIV Incidence Assays (14), and was used by major national surveys in southern Africa (15).

as well as in the discussion:

These figures provide important insights into ongoing transmission patterns in the region. However, they should be viewed critically since there is still a lack of proper field validation of these rapid recency assays (28) and especially misleading when not combined with VL testing (29). More research is needed to validate such assays in the field. 

15. Results

Could the authors revise how they present their results – for example the sentence “314 among them were HIV-positive on the day of enrolment and 257 of them were enrolled into VIBRA trial.” Starts with a number, refers to people as “them” and only present n and not also % - including %s would be helpful throughout – sometimes the authors present n, sometimes N, and sometimes % - please can the authors standardise and improve their presentation of results.

Response: We modified and hopefully improved the entire result presentation.

16. Figure 1 presents numbers with ‘ and not , - I have not seen this before in a paper and would suggest the authors follow generally accepted ways of presenting numbers.

Response: We adapted figure 1 accordingly.

17. Figure 1 does not address the issue raised in the abstract that some of the “newly” diagnosed people were aware of their status.

Response: This point has been addressed (see point 1) and is therefore now not applicable anymore.

18. Also be helpful to make clear the 6 VL <1000 transfer into the 225 non-recents in figure 2

Response: We adapted figure 2 accordingly.

19. The authors state that “Majority of the participants were from Mokhotlong district (216/243 [89%]) and 13/18 (72%) participants were located in the catchment area of the same 5 out of 19 facilities.” – as I don’t believe either the 5 or 19 facilities have previously been referred to, it is unclear what facilities the authors are referring to – please clarify

Response: Thank you for pointing this out. We deleted this misleading sentence and reduced the characteristics table to “district” only, since the inclusion of facilities does not add much valuable information but rather raises concerns in terms of confidentiality as outlined in your point 22 below.

20. The issue of 59 participants being aware of their status is surprising as they should have been excluded from the study sample at the start – the authors need to revise their analytical approach so these individuals are excluded from the start as they does not meet usual inclusion criteria for recency testing. I would also argue that 59 people knowing their status cannot be referred to as “only” given this is a surprisingly high number given they are supposed to be “newly” diagnosed (i.e. the number should be 0).

Response: We excluded them now from the start. VIBRA trial explicitly included newly diagnosed as well as known diagnoses and defaulters from care. It was a mistake on our side, we reworked now the entire analysis and sampling frame and hope that excluding them will clarify the overall message. We do note that others include such people in their “recency studies”, but we agree with the reviewer that in our case it does not make sense.

21. It is unclear why the authors include the 59 people who were aware of their HIV status in subsequent analysis – they need to be removed as they are not newly diagnosed and therefore do not qualify for recency testing.

Response: We kindly refer to point 20 above.

22. The authors present low values (including 1 and 0) by very specific and potentially defining variable categories in their table 1 – I personally would mask facility names and not present small values (instead adopting e.g. <5) – I feel how the data are presented is problematic.

Response: We agree with the reviewer and deleted this variable from the table.

23. Confidence intervals are presented alongside definitive results where both the n and the N are certain – could the authors please make clear how we are to interpret these confidence intervals.

Response: Currently, we only show confidence intervals in table 2, the risk factor analysis. In these analyses, we interpret the confidence intervals as showing no conclusive effect of any characteristic on recent outcome. And therefore we do not further interpret these intervals.

24. Discussion 

The first paragraph of the discussion is almost written like a bullet point summary of the results and presents the main findings again flatly – it would be helpful if the finding of being female being associated with recent infection was explained for example.

Response: We adapted the first paragraph and also added more explanation to the updated risk factor analysis.

25. Good to see the authors share limitations – these must also include the fact people who new their status were included in the sample and that a rapid test was used rather than a laboratory based LAg assay.

Response: We updated our limitation section and included the limitation of using a rapid point-of-care test instead of a lab-based assay. The other limitation has been resolved.

26. The discussion lacks a so-what – next steps – what this means. The discussion needs to be greatly strengthened.

Response: We restructured and reworked the entire discussion. We hope we could add and strengthen it to your expectation. 

Thank you for this careful review, this helped a lot in improving the manuscript. Although I understand that it is not a major publication, I do believe this data are worth publishing, I certainly learned a lot as part of this MSc project – and I hope we could improve the manuscript and discussion to be now a sound and comprehensive presentation of the findings.

---

## [Decision Letter · Decision Letter 1]

12 Oct 2022

PONE-D-22-02237R1Recent HIV infections among newly diagnosed individuals living with HIV in rural Lesotho: Secondary data from the VIBRA cluster-randomized trialPLOS ONE

Dear Dr. Mohloanyane,

Thank you for submitting your manuscript to PLOS ONE. After careful consideration, we feel that it has merit but does not fully meet PLOS ONE’s publication criteria as it currently stands. Therefore, we invite you to submit a revised version of the manuscript that addresses the points raised during the review process.

Thank you for providing comprehensive responses to the first round of reviewers' comments and for revising the manuscript appropriately. There are just a few small points raised by one reviewer - I hope if these can be addressed, we can move towards acceptance.==============================

We look forward to receiving your revised manuscript.

Kind regards,

Richard John Lessells, BSc, MBChB, MRCP, DTM&H, DipHIVMed, PhD

Academic Editor

PLOS ONE

Journal Requirements:

Reviewers' comments:

Reviewer's Responses to Questions

**Comments to the Author**

1. If the authors have adequately addressed your comments raised in a previous round of review and you feel that this manuscript is now acceptable for publication, you may indicate that here to bypass the “Comments to the Author” section, enter your conflict of interest statement in the “Confidential to Editor” section, and submit your "Accept" recommendation.

Reviewer #1: All comments have been addressed

Reviewer #2: All comments have been addressed

2. Is the manuscript technically sound, and do the data support the conclusions?

Reviewer #1: Partly

Reviewer #2: Yes

3. Has the statistical analysis been performed appropriately and rigorously? 

Reviewer #1: No

Reviewer #2: Yes

4. Have the authors made all data underlying the findings in their manuscript fully available?

Reviewer #1: No

Reviewer #2: No

5. Is the manuscript presented in an intelligible fashion and written in standard English?

Reviewer #1: Yes

Reviewer #2: Yes

6. Review Comments to the Author

Reviewer #1: Thank you for the opportunity to review a revised version of the manuscript. It reads much better than the previous version. However there are a few more minor issues that need to be addressed

Abstract

Line 29-30: the figure 184 should appear in the results unless the study went out to purposively enroll 184

Line 41-42: this sentence should say 7% of those who were newly diagnosed had acquired HIV in the preceding 6 months. It is difficult to follow as written at the moment

Introduction

Line 47- 48: should refer to targeted prevention interventions vs just targeted interventions

Results

Table 1: consider presenting viral loads on the log scale

Table 2 : in the column on recent infections , for the categorical n/N (%) should be presented for both test and reference groups i.e. for Biological sex it should be 11/108 (10.2) for females vs 2/76 (2.6) for males.

for district it should be 1/16 (6.3%) for Butha Buthe vs 12/168 (7.1%) for Mokhotlong

please correct this for categorical variables

Reviewer #2: My comments were addressed. There is no more critique.

My comments were addressed. There is no more critique.

7. PLOS authors have the option to publish the peer review history of their article (what does this mean?). If published, this will include your full peer review and any attached files.

Reviewer #1: **Yes: **Tendesayi Kufa

Reviewer #2: No

---

## [Author Response · Author response to Decision Letter 1]

20 Oct 2022

xxx point by point response letter uploaded separately xxx

Response to Reviewers

The recency of newly diagnosed HIV infections among the rural general population in Lesotho: Secondary data from the VIBRA cluster-randomized trial

Journal Requirements:

We have prepared our anonymized dataset and uploaded it onto Zenodo. The DOI is the following: 10.5281/zenodo.7230264 

We included the DOI link in our data sharing statement.

Direct link: https://zenodo.org/record/7230264#.Y1Fsj4LP3eo

Reviewer #1: 

Thank you for the opportunity to review a revised version of the manuscript. It reads much better than the previous version. However there are a few more minor issues that need to be addressed

Abstract

1) Line 29-30: the figure 184 should appear in the results unless the study went out to purposively enroll 184

Response: We have adapted the abstract accordingly:

Participants were recruited from August 2018 to May 2019 and 184 patient-samples included in this study.

2) Line 41-42: this sentence should say 7% of those who were newly diagnosed had acquired HIV in the preceding 6 months. It is difficult to follow as written at the moment

Response: We have adapted accordingly:

During door-to-door testing among a general population sample in rural Lesotho, 7% of those who were newly diagnosed had acquired HIV in the preceding 6 months.

Introduction

3) Line 47- 48: should refer to targeted prevention interventions vs just targeted interventions

Response: Thank you for pointing this out. We rephrased the sentence:

The identification of newly infected individuals is important to help HIV prevention programs to determine where interventions are most needed.

Results

4) Table 1: consider presenting viral loads on the log scale

Response: We changed the viral load to log scale.

5) Table 2 : in the column on recent infections, for the categorical n/N (%) should be presented for both test and reference groups i.e. for Biological sex it should be 11/108 (10.2) for females vs 2/76 (2.6) for males.

for district it should be 1/16 (6.3%) for Butha Buthe vs 12/168 (7.1%) for Mokhotlong

please correct this for categorical variables

Response: Thank you for this important point. We adapted the table accordingly.

Reviewer #2: 

My comments were addressed. There is no more critique.

Response: Thank you for your thorough review.

---

## [Editor Report · Decision Letter 2]

4 Nov 2022

Recent HIV infections among newly diagnosed individuals living with HIV in rural Lesotho: Secondary data from the VIBRA cluster-randomized trial

PONE-D-22-02237R2

Dear Dr. Mohloanyane,

We’re pleased to inform you that your manuscript has been judged scientifically suitable for publication and will be formally accepted for publication once it meets all outstanding technical requirements.

Kind regards,

Richard John Lessells, BSc, MBChB, MRCP, DTM&H, DipHIVMed, PhD

Academic Editor

PLOS ONE
---

## [Editor Report · Acceptance letter]

11 Nov 2022

PONE-D-22-02237R2 

Recent HIV infections among newly diagnosed individuals living with HIV in rural Lesotho: Secondary data from the VIBRA cluster-randomized trial 

Dear Dr. Mohloanyane:

I'm pleased to inform you that your manuscript has been deemed suitable for publication in PLOS ONE. Congratulations! Your manuscript is now with our production department. 

Kind regards, 

on behalf of

Dr. Richard John Lessells 

Academic Editor

PLOS ONE